# Vascular Adhesion Protein-1 (VAP-1) as Predictor of Radiographic Severity in Symptomatic Knee Osteoarthritis in the New York University Cohort

**DOI:** 10.3390/ijms20112642

**Published:** 2019-05-29

**Authors:** Eirini Bournazou, Jonathan Samuels, Hua Zhou, Svetlana Krasnokutsky, Jyoti Patel, Tianzhen Han, Jenny Bencardino, Leon Rybak, Steven B. Abramson, Uwe Junker, Karen S. Brown, Mukundan Attur

**Affiliations:** 1Roche Innovation Center New York, New York, NY 10016, USA; eirini.bournazou@roche.com (E.B.); brownks5@gmail.com (K.S.B.); 2Division of Rheumatology, Department of Medicine, NYU School of Medicine, New York, NY 10003, USA; jonathan.samuels@nyulangone.org (J.S.); Svetlana.Krasnokutskysamuels@nyumc.org (S.K.); jyoti.patel@nyulangone.org (J.P.); tianzhen.han@nyulangone.org (T.H.); stevenb.abramson@nyulangone.org (S.B.A.); 3Applied Bioinformatics Laboratories, NYU School of Medicine, New York, NY 10016, USA; hua.zhou@nyulangone.org; 4Department of Radiology, NYU Langone Medical Center/NYU School of Medicine, New York, NY 10016, USA; Jenny.Bencardino@nyumc.org (J.B.); leon.rybak@nyulangone.org (L.R.); 5Roche Innovation Center Basel, CH-4070 Basel, Switzerland; uwe.junker@roche.com

**Keywords:** osteoarthritis, radiographic severity, vascular adhesion protein-1, synovial fluid, immunohistochemistry, microarray, real-time polymerase chain reaction, Kellgren–Lawrence score

## Abstract

Background: To investigate the expression of vascular adhesion protein-1 (VAP-1) in joint tissues and serum in symptomatic knee osteoarthritis (SKOA) patients and examine whether VAP-1 levels predict increased risk of disease severity in a cross-sectional study. Methods: Baseline VAP-1 expression and soluble VAP-1 (sVAP-1) levels were assessed in the synovium synovial fluid and in the serum in cohorts of patients with tibiofemoral medial knee OA and healthy subjects. Standardized fixed-flexion poster anterior knee radiographs scored for Kellgren–Lawrence (KL) grade (0–4) and medial joint space width (JSW). KL1/2 vs. KL3/4 scores defined early and advanced radiographic severity, respectively. Biochemical markers assessed in serum or synovial fluids (SF) comprised sVAP-1, interleukin 1 receptor antagonist (IL-1Ra), interleukin 6 (IL-6), soluble receptor for advanced glycation end-products (sRAGE), C-C motif chemokine ligand 2 (CCL2), C-C motif chemokine ligand 4 (CCL4), cluster of differentiation 163 (CD163), high sensitivity C-reactive protein (hsCRP), and matrix metalloproteinases (MMPs)-1,-3,-9. Associations between biomarkers and radiographic severity KL1/2 vs. KL3/4 (logistic regression controlling for covariates) and pain (Spearman correlation) were evaluated. Results: Elevated levels of sVAP-1 observed in OA synovial fluid and VAP-1 expression in synovium based on immunohistochemical, microarray, and real-time quantitative polymerase chain reaction (qRT-PCR) analyses. However, serum sVAP-1 levels in OA patients were lower than in controls and inversely correlated with pain and inflammation markers (hsCRP and soluble RAGE). Soluble VAP-1 levels in serum were also lower in radiographically advanced (KL3/4) compared with early KL1/2 knee SKOA patients. Conclusion: Local (synovial fluid) semicarbazide-sensitive amine oxidase (SSAO)/sVAP-1 levels were elevated in OA and correlated with radiographic severity. However, systemic (serum) sVAP-1 levels were lower in SKOA patients than normal and inversely correlated with pain and inflammation markers. Serum sVAP-1 levels were higher in early (KL1/2) compared with advanced (KL3/4) SKOA patients.

## 1. Introduction

Osteoarthritis (OA), the most common type of arthritis, affects the whole joint and causes substantial disability in late life. On the basis of a recent report by the Center for Disease Control and Prevention (CDC), OA affects over 30 million US adults, and by 2040 it will affect an estimated 78 million (26%), with the national cost of arthritis being $304 billion overall [1]. OA is characterized by a breakdown of articular cartilage with other joint tissues (e.g., synovial membrane, subchondral bone, osteophytes) contributing to disease progression [2]. Although age, gender, and genetic predisposition are major intrinsic risk factors for the development of OA, the precise clinical and pathophysiological contributors to the disease incidence are unknown.

While formerly considered a non-inflammatory joint disease, inflammatory mediators produced by articular tissue are shown to be implicated in OA pathogenesis. Indeed, synovitis is associated with greater risk of cartilage loss, while cytokines and prostaglandins produced by cartilage promote cartilage degeneration [3,4]. Local inflammation within OA joint tissues has also been associated with increased levels of inflammatory proteins, such as IL-1β, interleukin 1 receptor antagonist (IL-1Ra), interleukin 6 (IL-6), monocyte chemoattractant protein-1 (MCP-1), Chemokine (C-X-C motif) ligand 9 (CXCL9 or MIG), vascular endothelial growth factor (VEGF) or granulocyte-macrophage-colony-stimulating factor (GM-CSF) in the sera of OA patients with joint effusions, compared with controls [5,6]. Elevated leukocyte IL-1β mRNA expression is also associated with increased knee pain [7], while plasma levels of IL-1Ra are modestly associated with the severity and progression of symptomatic knee OA (SKOA) in a causal fashion, independent of other risk factors excluding body mass index (BMI) [8].

Vascular adhesion protein-1 (VAP-1), encoded by the gene amine oxidase copper-containing 3 (*AOC3*), is a homodimeric type I membrane-bound adhesion protein that is abundantly expressed in mammalian vascular endothelial and smooth muscle cells, and adipocytes [9,10]. Under normal physiological conditions, endothelial cell VAP-1 is inactive due to sequestration in endosomes, but in response to local inflammatory signals, it rapidly translocates to the cell surface and acts to amplify the local inflammatory process. Cell surface VAP-1 promotes lymphocyte adhesion and transmigration through the endothelium into the inflamed tissue in concert with other leukocyte adhesion molecules [11]. Unlike other adhesion molecules, matrix-metalloproteases can cleave VAP-1 and generate a soluble form of VAP-1 (sVAP-1 or SSAO). Soluble VAP-1 acts as an ectoenzyme that catalyzes the oxidative deamination of primary amines into pro-inflammatory aldehydes in a reaction that also produces hydrogen peroxide and ammonium. The SSAO activity of VAP-1 contributes to the local inflammatory response via the localized upregulation of additional adhesion molecules, such as the selectins intercellular adhesion molecule-1 (ICAM-1) and vascular cell adhesion protein-1 (VCAM-1) [12,13].

VAP-1 is considered an attractive anti-inflammatory target because only cell surface VAP-1 in inflamed tissues is available for local pharmacological intervention by inhibitors [14], while VAP-1 in non-inflamed tissues is inaccessible due to its intracellular sequestration. Use of VAP-1 inhibitors demonstrated a reduction in the number of infiltrating leukocytes and inflammatory response in different models of inflammation [14,15]. In arthritis models, mice lacking VAP-1 showed attenuation of anti-collagen antibody-induced arthritis [15] and administration of VAP-1 inhibitors reduced clinical inflammation scores and disease progression (adjuvant-induced anti-type II collagen antibody-induced arthritis, and several other models of chronic arthritis) [15,16,17]. VAP-1 was also shown to be expressed in the synovia of patients with reactive arthritis (e.g., treatment-resistant Lyme arthritis) [18]; however, limited data are available in OA. Preliminary work by Filip et al. [19] recently indicated that VAP-1 is expressed in the chondrocytes of rat and human articular cartilage and its expression were significantly enhanced during the hypertrophic differentiation of chondrocytes along with an increase in matrix metalloproteinase 13 (MMP13) and alkaline phosphatase levels [19]. In humans, the same study also presented preliminary evidence suggesting that VAP-1 protein expression and SSAO activity are increased in human osteoarthritic cartilage relative to unaffected cartilage from the same donor [19]. A longitudinal case-control study identified serum sVAP-1 levels as a strong predictor of OA development even 10 years before radiographic evidence of the disease, increasing at compared with matched controls [20].

The purpose of the current study was to investigate VAP-1 (cell-associated) and its soluble form, sVAP-1 in synovial fluids and serum, in human OA. We examined the expression of VAP-1 in human OA tissues, and the association of serum/sVAP-1 levels with demographic characteristics, disease severity, and pain. We further correlated serum sVAP-1 levels with a panel of inflammatory markers in patients with SKOA at baseline. Our data indicate that VAP-1 and its soluble form, sVAP-1, are potential markers of OA expressed at early stages of the disease that correlate with age, gender, and OA-related pain and inflammation.

## 2. Results

### 2.1. Baseline Characteristics of the Subjects in the New York University (NYU) Cohort

Baseline demographic and clinical characteristics of the patients with SKOA and control subjects in the NYU cohort are shown in Table 1. Among patients with SKOA in the NYU cohort, 63% were female and 87% non-Hispanic. Mean ± standard deviation (SD) age was 61.21 ± 10.51 years, and in the non-OA control group was 54.63 ± 9.37 years, and 50% were female. The mean ± SD Western Ontario and McMaster Universities Osteoarthritis Index (WOMAC) [21] and visual analog scale (VAS) scores were 30.97 ± 24.73 and 46.97 ± 26.47, respectively. In the control group of >51-year old subjects (Table 2), the mean ± SD age was 60.79 ± 6.94 years, and 35.71% were female, while the OA cohort (>51 years of age) was 64.69 ± 8.34 years and 64.45% were female (Table 2).

### 2.2. Elevated SSAO/sVAP-1 Levels in Synovial Fluid of OA Patients

Baseline demographics and clinical characteristics of non-OA and OA subjects from whom synovial fluids and serum were collected are shown in Table 3. We first performed hyaluronidase treatment of synovial fluid to investigate the effects of synovial fluid viscosity on sVAP-1 level measurement by ELISA. The sVAP-1 levels measured were comparable in both treated and untreated samples. We used diluted synovial samples for all biomarker determinations.

Significant elevations in synovial fluid sVAP-1 concentrations were observed in the OA compared with the non-OA samples (Figure 1A). We further investigated whether the sVAP-1 levels in serum behave as surrogates and reflect knee synovial fluid concentrations. For this purpose, we performed Spearman rank correlation analyses between the matched synovial fluid and serum samples. We found a positive correlation between the levels of sVAP-1 in the synovial fluid and serum of OA patients (*r* = 0.47; *p* = 0.014). However, synovial fluid sVAP-1 levels (107.94 ± 41.42 ng/mL) were 2–4-fold lower than in serum (482.5 ± 132.5 ng/mL). Hence, local changes in sVAP-1 concentrations may have important pathophysiological implications for cartilage and synovium homeostasis. Furthermore, when OA patients segregated based on the stage of OA (Kellgren–Lawrence (KL) grade), we found that patients with KL4 had lower sVAP-1 (Figure 1B) levels. Conversely, synovial fluid sVAP-1 levels were higher in early KL1/2 than in advanced OA KL3/4 groups (121.5 ± 28.64 vs. 97.8 ± 44.6 ng/mL, respectively; *p* = 0.046) and positively correlated with IL-6-SF (synovial fluid) levels (*r* = 0.38; *p* = 0.012). In addition, C-C motif chemokine ligand 2 (CCL2)-SF levels were significantly elevated in normal controls than in OA patients (*p* = 0.004), whereas C-C motif chemokine ligand 4 (CCL4)-SF were higher in OA patients than in non-OA controls after adjustment for age, gender, and BMI (Table 3).

### 2.3. VAP-1 Locally Overexpressed in the Synovium and Not Cartilage of End-Stage Knee OA Patients

In light of the observed increase in sVAP-1 protein levels in the synovial fluid of OA patients, we investigated whether VAP-1 mRNA expression is also elevated in OA synovial tissue. As shown in Figure 1C,D by both microarray and real-time quantitative polymerase chain reaction (qRT-PCR) analysis, VAP-1 mRNA expression was significantly higher in the synovial tissue of OA patients than in non-OA controls. We next performed immunohistochemical (IHC) analysis to investigate VAP-1 expression in synovium from end-stage OA patients. Elevated expression of VAP-1 was observed along with the synoviocyte marker CD55 (Figure 1E). Parallel staining of OA-knee synovium tissue for CD31 and CD68 showed that VAP-1 expression localized around blood vessels, as expected based on VAP-1 function.

We further examined VAP-1 expression in OA cartilage tissues. The quantitative RT-PCR analysis of normal (*n* = 10) and OA (*n* = 11) cartilage for VAP-1 mRNA expression showed no difference in relative VAP-1 expression normalized to the housekeeping gene GAPDH (Figure 2A). Furthermore, immunohistochemical staining of VAP-1 in lesion relative to non-lesion knee cartilage from the same patient showed no significant difference (Figure 2B).

Serum sVAP-1 levels were lower in SKOA patients and inversely associated with radiographic severity. We determined the baseline serum sVAP-1 levels in SKOA (*n* = 372) and non-OA subjects (*n* = 23) from the NYU cohort. SKOA patients had significantly lower sVAP-1 levels than controls (430.00 ± 115.50 vs. 569.90 ± 178.01 ng/mL, respectively; *p* ≤ 0.002) and retained significance even after adjustment for covariates age, gender, and BM (Table 1 and Figure 3A). We next determined the correlation of serum sVAP-1 levels with radiographic KL grade and found that sVAP-1 levels were significantly lower in radiographically advanced KL4 grades (*p* = 0.022; Figure 3B). We further determined the association of serum sVAP-1 with radiographic severity and found that sVAP-1 were lower in KL3/4 than in KL1/2 (412.6 ± 102.7 vs. 437.6 ± 116.8 ng/mL, respectively; *p* = 0.02; Figure 3C) SKOA patients. Since OA prevalence increases significantly with age groups of ≥50 years [22,23], we next analyzed sVAP-1 levels by dichotomizing the sample into age groups of ≤51 and ≥51 years. Serum sVAP-1 levels were significantly higher in SKOA patients of ≥51 years than in patients ≤50 years of age (437.86 ± 111.13. vs. 396.50 ± 128.0 ng/mL, respectively; *p* = 0.009; Figure 3D).

Similarly, in the >51 years of age cohort (Figure 2D), serum sVAP-1 levels were significantly lower in radiographically advanced OA-KL3/4 than in early KL1/2 OA patients (415.6 ± 102.5 vs. 455.7 ± 111.3 ng/mL, respectively; *p* = 0.01). KL0 OA patients were excluded from the biomarker study.

### 2.4. Serum sVAP-1 Levels Correlated with Pain

After adjustment for age, gender, and BMI, serum sVAP-1 levels negatively correlated with baseline WOMAC pain (*r* = −0.16, *p* = 0.002) but not with baseline VAS pain (*r* = −0.16, *p* = 0.002). Conversely, we observed a positive association of both WOMAC and VAS pain with KL scores in SKOA patients (*r* = 0.19; *p* = 0.023; *r* = 0.175; *p* = 0.035, respectively).

### 2.5. Correlation of Baseline Serum sVAP-1 Levels with Other Biomarkers

In previous studies, we measured the levels of circulating biomarkers such as IL-1Ra, matrix metalloproteinase 1 (MMP1), matrix metalloproteinase 3 (MMP3), high sensitivity C-reactive protein (hsCRP), hyaluronic acid (HA), and soluble receptor for advanced glycation end products (sRAGE) in the NYU cohort. None of the biomarkers described above were differentially expressed between normal and SKOA patients. However, sVAP-1 levels were significantly lower in OA than in non-OA or normal subjects, even after adjustment for age, gender, and BMI (Table 2). In this NYU cohort, serum sVAP-1 levels also negatively correlated with hsCRP (*r* = −0.15; *p* = 0.006) and was consistently lower in the >51 years NYU OA cohort, which further displayed a negative correlation with hsCRP (*r* = −0.12; *p* = 0.05). Conversely, serum sVAP-1 levels did not correlate with any of the other biomarkers examined. These findings showed that serum sVAP-1SSAO levels are higher in early stages (KL1/2) than in advanced stages (KL3/4) of radiographic OA and display an opposite pattern of expression relative to inflammatory biomarkers.

## 3. Discussion

The results presented in this study show that elevation in serum and synovial VAP-1 and its soluble form, sVAP-1, and a correlation with radiographically less severe OA. In a cohort of patients with SKOA, a higher concentration of sVAP-1 levels was observed in early (KL1/2) than in advanced OA (KL3/4). In addition, our data also indicate that WOMAC (total) and VAS pain negatively correlate with serum sVAP-1 levels, whereas WOMAC and VAS moderately positively correlate with radiographic KL grades [24]. However, discordance exists between clinical/painful and radiographic knee OA [25].

It is known that OA is a disease of the whole joint, with an inflammatory component that affects cartilage, synovium, bone, ligament, and muscles, and leads to the deterioration and loss of joint function as the disease progresses [26]. Among the many molecular pathways activated within joint tissues during the course of OA are mediators that are classically associated with inflammation, such as IL-1β, TNF-α, nitric oxide, prostaglandins, and complement [2,27,28,29]. VAP-1 is a membrane-bound adhesion molecule that plays a key role in inflammation, as it regulates lymphocyte extravasation into the inflamed tissue [11]. Its soluble form, sVAP-1, possesses enzymatic activity, as it catalyzes the oxidative deamination of primary amines into pro-inflammatory aldehydes, hydrogen peroxide, and ammonium, thus amplifying the inflammatory signal [12,13]. Thus, synovial fluid sVAP-1, which remains at the site of local inflammation, may still play a role in amplifying the inflammatory cascade.

In this study, we show that VAP-1 is locally overexpressed in the synovium of end-stage knee OA patients, as determined by IHC analysis. VAP-1 overexpression correlates with increased expression of the synoviocyte markers CD55, CD68, and CD31, suggesting a complementary role of VAP-1 in local tissue inflammation. We confirmed these findings by microarray and qPCR analyses of the synovial tissue of OA patients. Our study is the first to report increased VAP-1 expression in the OA synovium and adds to a study by Filip et al. [19], which provided preliminary evidence of increased VAP-1 protein expression in human OA cartilage relative to undamaged cartilage from the same patient. Overall, our data confirm the existence of low-grade inflammation within the osteoarthritis joints and suggest VAP-1 may play a role in this inflammation.

Local inflammation within OA joint tissues reflected in serum biomarkers led to the understanding that there is evidence for systemic low grade inflammation in subsets of OA patients. Our laboratory has previously provided evidence of increased systemic levels of prostaglandin E2 (PGE2), interleukin-1β (IL-1β), IL-1 receptor antagonist (IL-1Ra) and matrix metalloproteinase-9 (MMP-9) in patients with SKOA [30]. Indeed, plasma PGE2 levels in SKOA patients were reported to be two-fold higher than in healthy controls and to correlate with disease severity, while plasma levels of IL-1Ra were modestly associated with the severity and progression of SKOA in a casual fashion, independently of other risk factors [8,29]. TSG-6 (TNF-stimulated gene-6), a hyaluronan-binding protein associated with inflammation, has also been identified as a promising independent biomarker for OA progression since no association was found between TSG-6 activities at baseline and four distinct OA progression states over a three-year period [4]. In this study, we have determined baseline serum sVAP-1 levels in SKOA patients and non-OA subjects. In the SKOA cohort, we further analyzed sVAP-1 levels by dichotomizing age groups into ≤50 and ≥51 years of age. Our results indicate that serum sVAP-1 levels were significantly higher in SKOA patients of ≥51 years of age than ≤51-year-old patients and negatively correlated with radiographic severity based on KL grade, as serum sVAP-1 levels were significantly lower in KL3/4 OA than in KL1/2 patients. We observed a similar negative correlation between serum sVAP-1 levels with baseline WOMAC pain and hsCRP and no correlation with other biomarkers displaying in this way, an opposite pattern of expression relative to inflammatory biomarkers, therefore indicating an association between high serum levels of sVAP-1 in early stages of OA. In addition, serum sVAP-1 levels also negatively correlated with CRP levels in prediabetes patients [31].

With respect to synovial fluid analyses, it is of interest that local sVAP-1 levels were higher in OA patients than in healthy controls. However, higher serum sVAP-1 levels in early knee OA patients could be a surrogate marker for less severe radiographic OA.

So far, several studies have identified many candidate molecular biomarkers from various joint tissues, such as markers of extracellular matrix synthesis or degradation, whereas other markers are inflammatory in nature. We propose sVAP-1 levels as a predictor of OA development that can be used, which may help to identify patients at increased risk of developing radiographic severity, and can ultimately facilitate the development of disease-modifying OA drugs (DMOADs).

## 4. Materials and Methods

### 4.1. Symptomatic Knee OA (SKOA), New York University (NYU) Patient Cohort

We assessed a subset of 372 individuals with symptomatic knee OA and 24 non-OA control subjects enrolled in a cross-sectional study as part of an NIH-funded (R01-AR052873), natural history knee OA study with a focus on inflammatory biomarker discovery. At initial enrollment, subjects met clinical ACR criteria for knee OA and had symptomatic OA of at least one knee (index knee), with a Kellgren–Lawrence (KL) score ≥1 in that knee. Exclusion criteria included any other form of arthritis (e.g., rheumatoid arthritis, spondyloarthritis, gout, pyrophosphate disease, or other crystal arthropathy), BMI ≥ 33 kg/m^2^ (chosen to minimize the impact of obesity on OA/inflammatory biomarkers, without adversely impairing recruitment potential), and other characteristics as previously described [8,28]; one of two investigators (SK, JS) examined all patients. Radiographic assessments at baseline included bilateral (signal and non-signal knee) KL determination and quantitative measurement of joint space width (JSW), all performed by musculoskeletal radiologists (JB, LR) blinded to patient information. Between the readers, x- a Kappas for the inter-rater agreement was 0.85 for KL scores and Kappas for JSW were ≥0.93 for medial compartments of knees. Non-OA control subjects had no pain, and no clinical signs or symptoms of arthritis. The Western Ontario and McMaster Universities Osteoarthritis Index (WOMAC) and a visual analog scale (VAS) for pain assessment [7], assessed non-fasting blood samples from subjects in the NYU cohort collected in serum collection tubes and in heparinized pyrogen-free tubes (for plasma; BD Biosciences). Blood was processed within 30–60 min of the collection as described [8]. Plasma and serum samples were aliquoted and stored at −70 °C until tested. The institutional review board (IRB) of the NYU School of Medicine approved this study (# i05-131 and i12-03682, 20 February 2019 and 3 December 2018 respectively). 

### 4.2. Synovial Fluid, Synovium, and Cartilage from Non-OA Subjects and OA Patients

Tissue samples were obtained from a few different cohorts: (a) Normal human synovial fluid and synovium were collected at autopsy (4–24 h after death), procured from the National Disease Research Interchange (NDRI) and stored frozen at −80 °C. The NDRI collected extensive health information, and no clinical diagnosis of OA was reported for these subjects. However, knee X-ray screening data were not available.

(b) OA synovium and cartilage was collected from end-stage knee OA patients undergoing knee replacement surgery at NYU Langone Orthopedic Hospital. Their use in the de-identified form in the current study was in accordance with the ethical standards of the Helsinki Declaration of 1975, as revised in 2000, and was approved (#i9018, 23 January 2019) by the institutional review board (IRB) of the NYU School of Medicine.

(C) OA synovial fluid and serum samples were collected as part of an observational study to determine factors that influence knee OA pain improvement with hyaluronic acid visco-supplementation. A subset of OA patients from this cohort had knee effusions, and standard-of-care included aspiration of the effusions before hyaluronic acid (HA) injection, all performed by the orthopedic surgeons and rheumatologists. The synovial fluid samples were collected without joint lavage, and the volume ranged from 0.5 to 30 mL. The fluid cell-free supernatants were prepared and frozen (−80 °C) within one hour of collection. Collection and storage of synovial fluids were approved (#13-01257, 21 May 2019) by the IRB of the NYU School of Medicine.

Later, the synovial fluid was diluted at least 1:1 with the assay diluent of the respective ELISA kits, and fluid markers were determined using the relevant ELISA kits (IL-8 (D8000C); high-sensitivity IL-6 (HS600B) and IL-10 HS100C); CCL2 (DCO00); CCL4 (DMB00); and VAP-1/AOC3 (DVAP10)), as recommended by the manufacturer (R&D Systems, Minneapolis, MN, USA.

### 4.3. Microarray Gene Expression Analysis

For first strand cDNA synthesis, five micrograms of total RNA were used for generating complementary RNA for microarray hybridization using an Enzo kit (Affymetrix, Santa Clara, CA, USA) and purified using the Qiagen RNeasy kit (Qiagen, Waltham, MA, USA). The complementary RNA was fragmented at 95 °C for 35 min for target preparation to hybridize against the Human Genome U133A microarray (Affymetrix). Raw expression data from array scans were pre-processed and normalized using the robust multichip average (RMA) method as described [7,32,33].

### 4.4. Cytokine and sVAP-1 Measurement

Plasma protein biomarkers (cluster of differentiation 163 (CD163), MMP-1, -3, -9) were analyzed using Meso Scale Discovery (MSD) and IL-1Ra kits (Cat# DRA00B R&D systems, Minneapolis, MN, USA). Hyaluronic acid (HA) was measured by enzyme-linked binding protein assay (Corgenix, Broomfield, CO, USA) and high-sensitivity C-reactive protein (hsCRP) by ELISA (07BC-1119, MP Biomedicals, Solon, OH, USA). Soluble serum SSAO/sVAP-1 levels were measured by ELISA following the manufacturer’s instructions (R&D Systems, Minneapolis, MN, USA). All biomarkers were in measurable ranges; no imputation was required for statistical analyses.

### 4.5. Real-Time Quantitative Polymerase Chain Reaction (qRT-PCR)

Total RNA was purified from human normal, and OA synovium or cartilage as described [29,30], and its quality was assessed by measurement of optical density at 260 and 280 nm. One microgram of total RNA was primed with oligo (dT) 18 primers, and complementary DNA was synthesized using a cDNA Synthesis Kit (Clontech, Mountain View, CA, USA). Predesigned TaqMan primer sets—AOC3 or VAP-1 (Hs00907292_m1) and GAPDH (Hs99999995_m1) (Thermo Fisher Scientific, Rockford, IL, USA) were used, and target (SSAO) mRNA expression was normalized to the housekeeping gene GAPDH. qRT-PCR reactions were run in an ABI Prism 7300 sequence detection system (Thermo Fisher Scientific, Rockford, IL, USA). Relative gene expression levels were calculated using the 2^∆∆*C*t^ method [34].

### 4.6. Histological and Immunohistochemical Analysis of Human Synovial Tissue and Cartilage

Freshly isolated human normal and OA knee synovial tissue samples (*n* = 4) were immediately fixed in 4% paraformaldehyde for 24 h, transferred to 70% ethanol and embedded in paraffin. Cartilage specimens were harvested and fixed in 4% paraformaldehyde for 24 h, decalcified in 10% ethylenediaminetetraacetic acid (EDTA) for 4 weeks, and embedded in paraffin. Five-micrometer sections were cut, deparaffinized, rehydrated and stained with primary antibodies (anti-VAP-1, -CD55, -CD31, -CD68, -synovium and -C2M, -ARGxx, anti-VAP-1-cartilage) and the Vectastain ABC kit (Vector Laboratories, Burlingame, CA, USA), according to the manufacturer’s instructions. The following antibodies were used: Sigma (Ronkonkoma, NY, USA): VAP-1 (#HPA000980), LS Biosciences (Seattle, WA, USA), CD55 (#LS-C123072); Abcam (Cambridge, MA, USA): CD31 (#ab9498)and CD68 (#790-2931) Ventana (Tucson, AZ, USA). C2M (collagen fragment) and ARGxx (aggrecan fragment) antibodies were generously provided by Dr. Anne-Christine Bay-Jensen (Nordic Biosciences, Herlevc, Norway). Anti-IgG antibody alone was used as a negative control. All histology and immunohistochemistry were performed at the Experimental Pathology Research Laboratory of New York University Langone Medical Center.

### 4.7. Statistical Analysis

OA stages were defined by KL grade (1–4) and dichotomized (KL1/2 vs. KL3/4) for statistical analyses. Baseline SSAO/sVAP-1 was assessed for association with baseline radiographic severity (early KL1/2 vs. advanced KL3/4) in SKOA subjects. Potential confounding variables are known to be associated with OA? were included in the models (age, gender, BMI). Data were expressed as means + SD. All statistical analyses were performed with Graph Pad Prism 7.03 (San Diego, CA, USA). One-way analysis of variance (ANOVA) was used to test the statistically significant differences among KL groups. A nonparametric two-tailed Mann–Whitney was used for pairwise comparisons between groups, and Spearman’s correlation coefficient was used for correlation analyses.

### 4.8. Ethics Approval

The current study was in accordance with the ethical standards of the Helsinki Declaration of 1975, as revised in 2000, and the studies (#i9018, i05-131, i12-03682, and 13-01257, 23 January 2019, 20 February 2019, 3 December 2018, and 21 May 2019 respectively) were approved by the institutional review board (IRB) of the NYU School of Medicine. All patients provided written, informed consent to participate in the study.

## 5. Conclusions

Local (synovial fluid) SSAO/sVAP-1 levels were elevated in OA and correlated with radiographic severity. However, systemic (serum) sVAP-1 levels were lower in SKOA patients than normal and inversely correlated with pain and inflammation markers. Serum sVAP-1 levels were higher in early (KL1/2) compared with advanced (KL3/4) radiographic knee SKOA patients. A limitation of this study is the inability to compare serum, OA synovial fluid, and synovium specimens from the same subjects, and therefore it is difficult to determine whether local VAP-1 expression and sVAP-1 levels in OA synovium correlate with disease severity and progression. Another factor that limits our conclusions is that VAP-1 differential expression in the joint tissues may reflect processes occurring more globally in the patient besides synovium and cartilage. In addition, our synovial sample of 45 OA subjects was still rather small but had the advantage of containing well-characterized OA patients. Finally, because of the non-availability of knee radiographic data for the normal controls used in the synovial fluid study, we cannot exclude the possibility that some of the controls had asymptomatic knee OA. The current findings need validation in other, larger prospective patient populations studies. However, our data suggest that local VAP-1 expression at early stages of the disease may act to promote and maintain inflammation.

## Figures and Tables

**Figure 1 ijms-20-02642-f001:**
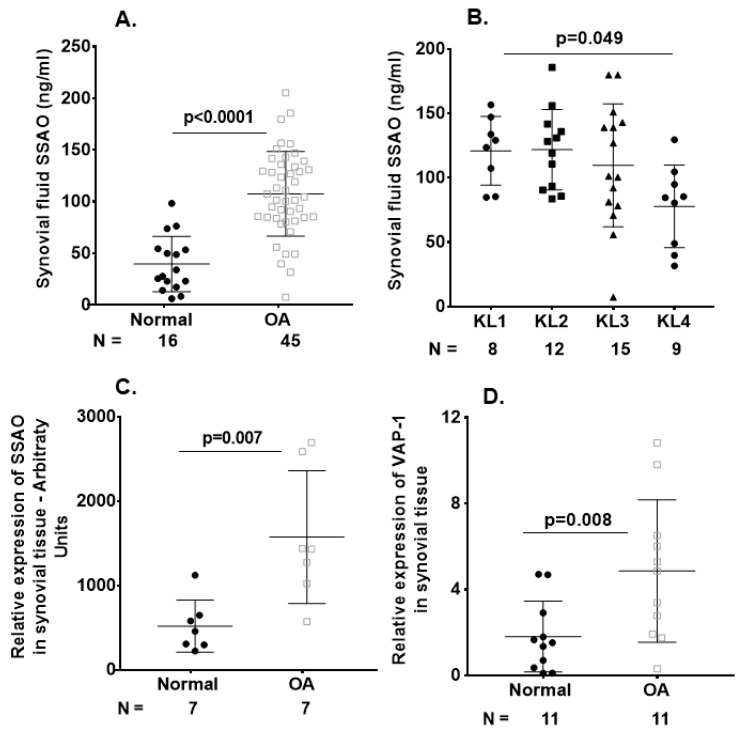
Vascular adhesion protein-1 (VAP-1) levels were elevated in osteoarthritis (OA) synovium and sVAP-1 levels in OA synovial fluid. (**A**) ELISA of sVAP-1 in the synovial fluid of control (normal) and OA patients; (**B**) Comparison of synovial fluid sVAP-1 levels in radiographic Kellgren–Lawrence (KL) (KL 1–4) groups in OA patients; (**C**) Microarray analysis of VAP-1 expression in synovial tissue. Total RNA from seven normal and OA synovia was hybridized to human U133A arrays. After normalization, VAP-1 expression was shown in arbitrary units; (**D**) Real-time quantitative polymerase chain reaction (qRT-PCR) analysis of VAP-1 mRNA in 11 normal and 11 OA synovium specimens from knee joints; (**E**) Immunohistochemical analysis of VAP-1 in normal and end-stage OA knee synovium. OA knee synovium sectioned and stained for VAP-1 with vectastain reagents (Vector Laboratories, Burlingame, CA, USA). Representative H&E, VAP-1, CD55, CD31, and CD68 images are shown. Leica SCN400F scanned the whole slide at bright-light and 40× magnification (0.25 μm/pixel). The images are shown with scale bar 100 μm and 10 μm. For synovium tissue slides stained with H&E and VAP-1, positive cells were counted using magnified photos on four sections per specimen, and cell numbers were averaged by each sample. CD55, CD31, and CD68 immunohistochemical staining were shown as a percentage of the staining area in the region of interest. Photos were analyzed with Image J by converting RGB (red, green, and blue) photos into binary images, and the stained areas were lined out and analyzed. One-way analysis of variance (ANOVA) (**D**) determined *p* values or nonparametric Mann–Whitney test between control and OA groups (**A**,**C**,**D**). Symbols represent individual subjects; bars show mean and standard deviation. *N*: number of individuals whose samples were tested.

**Figure 2 ijms-20-02642-f002:**
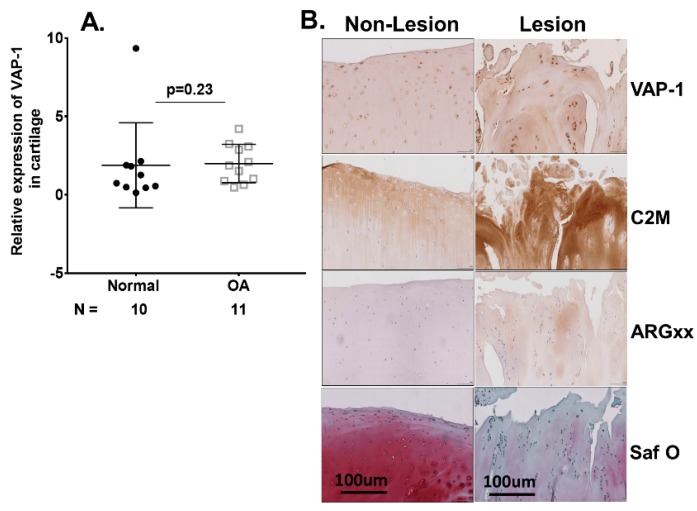
(**A**) RT-qPCR analysis of VAP-1 mRNA in 10 normal and 11 OA individual cartilage specimens from knee joints. Symbols represent individual subjects; bars show the mean and standard deviation. *N* values are the number of individuals from whom samples were available for testing. *p* values were determined by nonparametric Mann–Whitney test; (**B**) Immunohistochemical analysis of VAP-1 in lesional and non-lesional OA knee cartilage. Lesion and non-lesion OA knee cartilage samples (*n* = 3) from same individual undergoing total knee replacement surgery were collected, sectioned, and stained for VAP-1 with vectastain reagents (Vector Laboratories, Burlingame, CA, USA). Representative VAP-1, C2M, and ARGxx immunostaining images are shown. Slides were scanned at 40× and micrographs were taken at a magnification of 20×. Safranin O staining showed loss of extracellular matrix and cartilage degeneration in lesion compared with the non-lesion area of the same patient. *p* values were calculated by the nonparametric Mann–Whitney *U* test between the control and OA groups. The images are shown with scale bar 100 μm. *N*: number of individuals whose samples were tested.

**Figure 3 ijms-20-02642-f003:**
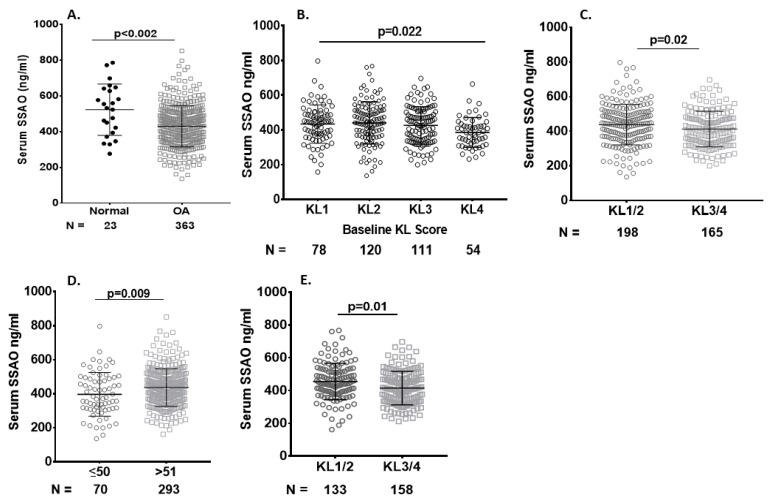
Serum levels of sVAP-1 are lower in OA patients than in controls. (**A**) Lower baseline serum sVAP-1 levels among patients with SKOA compared to normal. Baseline serum sVAP-1 levels ranged from 276.32 to 1105.39 ng/mL in non-OA controls and from 136.16 to 851.32.2 ng/mL in SKOA patients; (**B**) Comparison of serum sVAP-1 levels in radiographic KL (KL 1–4) groups of SKOA patients; (**C**) Lower serum sVAP-1 (412.6 + 102.7 ng/mL) levels in advanced radiographic (KL3/4) than early (KL1/2) 437.6 + 116.6 ng/mL SKOA patients; (**D**) Soluble VAP-1 levels were significantly higher in the age group >51 years than in ≤50 year-old SKOA patients. Soluble VAP-1 levels ranged from 136.16 to 796.6 ng/mL in <50 years and from 136.16 to 796.6 ng/mL in > 51-year-old SKOA patients; (**E**) Serum sVAP-1 levels were also lower in advanced (KL3/4) than early (KL1/2) compared with the general age group as shown above. *N*: number of individuals whose samples were tested. *p* values determined by nonparametric Mann–Whitney test or one-way analysis of variance (ANOVA). Bars in the graph shows the mean and standard deviation.

**Table 1 ijms-20-02642-t001:** Baseline demographics and biomarker levels of the New York University (NYU) cohort of normal subjects and symptomatic knee osteoarthritis (SKOA) patients.

Variable	Normal (*N* = 24)	SKOA (*N* = 372)	*p* Values	Adjusted for AGB *p* Values
Age (years)	54.63 (9.37), [40.00–75.00]	61.21 (10.51), [37.00–88.00]	**0.003**	
Gender (%)				
Male	50%	36.93%		
Female	50%	63.07%		
BMI (kg/m^2^)	26.30 (3.92), [20.00–32.40]	26.35 (3.50), [15.40–32.60]	0.943	0.943
VAS (0–100)	0	46.97 (26.47), [0.00–100.00]	NA	NA
WOMAC (0–100)	0	30.97 (24.73), [0.00–98.60]	NA	NA
JSW (mm)	NA	3.39 (1.34), [0.00–7.30]	NA	NA
KL Grades				
KL0	63.64%	5.41%		
KL1	36.36%	15.68%		
KL2	0.00%	32.70%		
KL3	0.00%	31.35%		
KL4	0.00%	14.86%		
IL-1Ra (pg/mL)	354.53 (172.28), [91.40–726.43]	350.04 (156.37), [97.97–999.98]	0.901	0.943
MMP-1 (pg/mL)	2388.24 (1140.35), [1137.52–4679.16]	3151.94 (2383.33), [401.28–17,537.19]	0.146	0.351
MMP-3 (pg/mL)	16,525.13 (8777.40), [6684.15–33,139.95]	17,517.27 (16,424.86), [3768.51–244,674.57]	0.784	0.943
MMP-9 (ng/mL)	49,197.66 (56,813.60), [9256.92–257,706.04]	38,409.48 (29,838.25), [9426.51–197,065.54]	0.135	0.351
IL-15 (pg/mL)	1.18 (0.30), [0.70–1.90]	1.15 (0.32), [0.6–3.29]	0.714	0.943
hsCRP (mg/L)	4.06 (4.85), [0.14–19.85]	2.73 (3.68), [0.00–37.00]	0.117	0.351
CD163 (ng/mL)	679.84 (317.42), [291.74–1852.40]	656.09 (236.68), [164.69–1501.75]	0.692	0.943
HA (ng/mL)	29.59 (18.07), [1.91–72.72]	30.20 (29.75), [0.76–193.22]	0.929	0.943
sRAGE (ng/mL)	1303.70 (542.76), [435.92–2463.60]	1106.13 (571.74), [239.11–4657.23]	0.182	0.363
sVAP-1 (ng/mL)	569.90 (178.01), [276.32–1105.39]	430.00 (115.50), [136.16–851.32]	**<0.002**	**<0.0001**

Data shown are mean and standard deviation (SD), with minimum and maximum levels shown in brackets. BMI = body mass index; VAS = visual analog scale; WOMAC = Western Ontario and McMaster Universities Osteoarthritis Index; JSW = joint space width; KL = Kellgren–Lawrence score; IL-1Ra = interleukin 1 receptor antagonist; MMP-1, -3, -9 = matrix metalloproteinase -1, -3, or -9, respectively; IL-15 = interleukin 15; hsCRP = high sensitivity C-reactive protein; CD163 = cluster of differentiation 163 (macrophage marker); HA = hyaluronic acid; sRAGE = soluble receptor for advanced glycation end-products; sVAP-1: soluble vascular adhesion protein-1; AGB = age, gender, and BMI; NA = not available; significant *p* values are bold typed.

**Table 2 ijms-20-02642-t002:** Baseline demographics and biomarker levels of normal controls and SKOA patients of ≥51 years of age.

Variable	Normal (*N* = 14)	SKOA (*N* = 301) ≥51 Years	*p* Values	Adjusted for AGB *p* Values
Age (years)	60.79 (6.94), [51.00–75.00]	64.69 (8.34), [51.00–88.00]	0.086	0.305
Gender (%)				
Male	64.29%	35.55%		
Female	35.71%	64.45%		
BMI (kg/m^2^)	26.62 (4.08), [20.00–31.90]	26.35 (3.52), [15.40–32.60]	0.779	0.934
VAS (0–100)	0	46.12 (25.83), [0.00–100.00]	NA	NA
WOMAC (0–100)	0	30.66 (23.25), [0.00–91.60]	NA	NA
JSW (mm)	NA	3.28 (1.37), [0.00–7.30]	NA	NA
KL Grades				
KL0	46.2%	4.3%		
KL1	53.8%	12.6%		
KL2	0.0%	30.6%		
Kl3	0.0%	34.6%		
KL4	0.0%	17.9%		
IL-1Ra (pg/mL)	362.21 (163.82), [91.40–702.70]	363.47 (154.69), [97.97–999.98]	0.977	0.977
MMP-1 (pg/mL)	2511.82 (1290.26), [1137.52–4679.16]	3230.39 (2430.58), [401.28–17,537.19]	0.274	0.657
MMP-3 (pg/mL)	15,704.52 (7796.96), [6684.15–33,139.95]	17,546.72 (17,564.45), [4685.58–244,674.57]	0.697	0.934
MMP-9 (ng/mL)	41,278.95 (36,076.83), [9256.92–150,916.05]	38,662.00 (28,912.33), [9426.51–184,835.39]	0.745	0.934
IL-15 (pg/mL)	1.28 (0.28), [0.70–1.90]	1.14 (0.30), [0.63–3.29]	0.102	0.305
hsCRP (mg/L)	4.80 (5.59), [0.14–19.85]	2.91 (3.96), [0.00–37.00]	0.090	0.305
CD163 (ng/mL)	749.40 (348.24), [483.96–1852.40]	687.27 (235.73), [223.48–1501.75]	0.396	0.680
HA (ng/mL)	34.51 (17.67), [13.25–72.72]	33.78 (31.59), [0.76–193.22]	0.933	0.977
sRAGE (pg/mL)	1342.65 (492.87), [802.28–1927.11]	1142.09 (615.22), [293.75–4657.23]	0.396	0.680
sVAP-1 (ng/mL)	577.93 (192.95), [332.83–1105.39]	437.86 (111.13), [161.94–851.32]	**<0.0001**	**<0.0001**

Data shown are mean and standard deviation (SD), with minimum and maximum levels of each biomarker shown in brackets. BMI = body mass index; VAS = visual analog scale; WOMAC = Western Ontario and McMaster Universities Osteoarthritis Index; JSW = joint space width; KL = Kellgren–Lawrence score; IL-1Ra = interleukin 1 receptor antagonist; MMP-1, -3 -9 = matrix metalloproteinase -1, -3, or -9, respectively; IL-15= interleukin 15; hsCRP = high sensitivity C-reactive protein;; CD163 = cluster of differentiation 163 (macrophage marker); HA = hyaluronic acid; sRAGE = soluble receptor for advanced glycation end-products; AGB = age, gender, and BMI; NA = not available; significant *p* values are bold typed.

**Table 3 ijms-20-02642-t003:** Baseline demographics and biomarker levels of synovial fluid donors.

Variable	Normal (*N* = 20)	OA (*N* = 45)	*p* Values	Adjusted for AGB *p* Values
Age (years)	75.00 (10.72), [51.00–90.00]	62.15 (11.47), [43.00–87.00]	**<0.00001**	
Gender (%)				
Male	66.67%	45.83%		
Female	33.33%	54.17%		
BMI	NA	28.74 (5.08), [20.00–44.09]	NA	NA
KL Grades				
KL1	NA	17%		
KL2	NA	26%		
KL3	NA	33%		
KL4	NA	20%		
KL not known		4%		
IL-8 (pg/mL)	22.38 (39.90), [3.21–134.79]	24.68 (42.13), [1.04–249.18]	0.839	0.979
HSIL-6 (pg/mL)	122.62 (193.02), [0.3–496.45]	122.22 (140.02), [5.26–630.25]	0.993	0.993
CCL2 (pg/mL)	682.53 (631.01), [0.10–2125.20]	357.34 (179.17), [36.73–953.08]	**0.002**	**0.004**
CCL4 (pg/mL)	1.12 (4.43), [0.10–19.39]	25.49 (38.31), [0.10–154.01]	**0.008**	**0.013**
HSIL-10 (pg/mL)	1.41 (2.84), [0.14–12.17]	1.03 (2.40), [0.05–16.27]	0.597	0.835
sRAGE (pg/mL)	146.8 (89.63), [13.38–327.40]	232.22 (163.7), [10.8–970.00]	**0.041**	**0.046**
SF-sVAP-1 (ng/mL)	38.12 (22.98), [6.20–88.35]	107.94 (41.42), [7.38–05.47]	**<0.0001**	**<0.0001**
sVAP-1 (ng/mL)—serum	Not available	482.5 (132.5), [180.6–695.6]		

Data shown are mean and standard deviation (SD), and minimum and maximum levels of each biomarker are shown in brackets. OA = osteoarthritis; BMI = body mass index; KL = Kellgren–Lawrence score; IL-8 = interleukin 8; HSIL-6 = high sensitivity interleukin 6; CCL2 = C-C motif chemokine ligand 2; CCL4 = C-C motif chemokine ligand 4; HSIL-10 = high sensitivity interleukin 10; SF = synovial fluids; AGB = age, gender, and BMI; NA = not available; significant *p* values are bold typed.

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
