# Peer review of "Vascular Adhesion Protein-1 (VAP-1) as Predictor of Radiographic Severity in Symptomatic Knee Osteoarthritis in the New York University Cohort"

_ijms, 2019, doi:10.3390/ijms20112642_

Round 1

Reviewer 1 Report

The authors investigate the role of VAP1 as an indicator severity in SKOA. The authors show a correlation between VAP1, however no mechanistic data are presented. The immunostaining is unclear since the data do not show similar regions within the tissue. Moreover the comparison between normal and OA cannot be concluded (figure 2). The authors have 5 patients with a large variability. Therefore they either need more patients, remove outliers or dismiss the data.

Author Response

Reviewer 1

Comments and Suggestions for Authors

The authors investigate the role of VAP1 as an indicator of severity in SKOA. The authors show a correlation between VAP1. However no mechanistic data are presented. The immunostaining is unclear since the data do not show similar regions within the tissue. Moreover, the comparison between normal and OA cannot be concluded (figure 2). The authors have 5 patients with large variability. Therefore they either need more patients, remove outliers or dismiss the data.

Response:  We agree with the reviewer that our current study does not present mechanistic data. However, we are pursuing mechanistic studies to understand the role of VAP-1 in animal models of OA, and we hope to report our findings in a future publication. We agree with the reviewer that the immunostaining does not show the same regions within the tissue. We tried our best to immunostain serial sections of the same region, but we could not obtain consecutive sections of high enough quality for staining the various markers. Unfortunately, the surgical material, especially the normal tissue, was limited and this prevented the execution of immunostaining of the same regions for any given patient. As suggested by the reviewer, we removed Figure 2A due to the large variability in VAP-1 mRNA expression in human cartilage samples, and we do not have additional microarray data to increase the number of samples in each group.

Reviewer 2 Report

The revised manuscript addressed the comments well. Some minor errors need to be corrected. This manuscript can be considered for publication.

line 157  97.8 + 44.6 should be corrected to 97.8 ± 44.6

line 172 should delete 40x, 20x, and indicate the bar length with bar in the photos, because the photo can be magnified on the screen, and thus was not 40x or 20x magnified as indicated

line 173 should delete 40x

line 208 should delete 20x, and add bar and bar length in photo

Author Response

Reviewer 2

The revised manuscript addressed the comments well. Some minor errors need to be corrected. This manuscript can be considered for publication.

line 157  97.8 + 44.6 should be corrected to 97.8 ± 44.6

line 172 should delete 40x, 20x, and indicate the bar length with a bar in the photos, because the photo can be magnified on the screen, and thus was not 40x or 20x magnified as indicated

line 173 should delete 40x

line 208 should delete 20x, and add bar and bar length in the photo

Response:  We thank the reviewer for highlighting the errors. We have revised the manuscript accordingly. 

Round 2

Reviewer 1 Report

Since the authors cannot include any mechanistic data the data are just correlative and therefore do not really establish a rational for the study. The study is purely observational and therefore very limited in its findings.

This manuscript is a resubmission of an earlier submission. The following is a list of the peer review reports and author responses from that submission.

Round 1

Reviewer 1 Report

The purpose of the  study was to investigate VAP-1  and its soluble form,
97 sVAP-1 in synovial fluids and serum, in human OA. The authors examined serum and synovial fluid for expression of markers for OA and inflammation, however, the should include a rational why they selected the specific markers and cytokines and MMPs. Why did they not choose COMP levels that are known to be different in OA patients and other markers? Figure 1, the authors should provide a negative control for the tissue staining. Based on the image the expression in the synovium tissue is way higher then 8 times for VAP1. They should show a reflective image. Also why did they not quantify the expression of the other markers? It seems the regions the images were taken in the synovial tissue are different in the different markers. Can the authors also include some more high magnification images?

Figure 2. It does not seem that there is more VAP1 sites, it seems the staining of the background is enhanced in the lesion and therefore also the VAP-1. Can the authors provide images that show their conclusion? I don't agree with the conclusion that VAP is locally overexpressed. There is no change in the qPCR and the RNA levels.Or is the heading wrong: VAP-1 locally overexpressed in the synovium of end-stage knee OA patients.

More details on the quantification of the IHF is needed and higher resolution images. The primers for the qPCR should be included. Apropriate negative controls are needed.More smaples of the OA patients and less of normal patients were analyzed. Can this effect the statistical analysis and maybe the authors need to use a different statistical test.

Reviewer 2 Report

Summary:

This manuscript reported a study aiming to investigate the expression of vascular adhesion protein -1 (VAP-1) in joint tissues and serum in symptomatic knee osteoarthritis (SKOA) patients and examine whether VAP-1 levels predict increased risk of disease severity in a cross-sectional study. The authors assessed baseline VAP-1 expression and soluble VAP-1 (sVAP-1) levels were assessed in the synovium synovial fluid and in the serum in cohorts of patients with tibiofemoral medial knee OA and healthy subjects. The authors measured Kellgren Lawrence (KL) grade (0–4) and medial joint space width (JSW) to define KL1/2 vs. KL3/4 scores; biochemical markers assessed in serum or synovial fluids (SF) including sVAP-1, IL-1Ra, IL-6, sRAGE, CCL2, CCL4, CD163, hsCRP, and MMPs-1,-3,-9. Then they evaluated associations between biomarkers and radiographic severity KL1/2 vs. KL3/4 (logistic regression controlling for covariates) and pain (Spearman correlation). The results showed that elevated levels of sVAP-1 observed in OA synovial fluid and VAP-1 expression in synovium based on immunohistochemical, microarray and qRT-PCR analyses. However, serum sVAP-1 levels in OA patients were lower than in controls and inversely correlated with pain and inflammation markers (hsCRP and soluble RAGE). Soluble VAP-1 levels in serum were also lower in radiographically advanced (KL3/4) compared to early KL1/2 knee SKOA patients. The authors concluded that local (synovial fluid) SSAO/sVAP-1 levels were elevated in OA and correlate with radiographic severity. However, systemic (serum) sVAP-1 levels lower in SKOA patients than normal and inversely correlated with pain and inflammation marker. Serum sVAP-1 levels were higher in early (KL1/2) compared to advanced (KL3/4) SKOA patients.

Comments:

This study showed some evidence that local (synovial fluid) SSAO/sVAP-1 levels related to KOA. However, discordance existed that systemic (serum) sVAP-1 levels lower in SKOA patients than normal and inversely correlated with pain and inflammation marker. Such a situation needed more clarification about the real mechanism. It seemed very important to select the good controls in this study. Other issues also need to be addressed.

1. Since knee OA is multi-factorial and symptoms may vary among patients, the authors need to present more the clinical histories. For example, in Table 1. The Age (years) of Normal (N=24) was 54.63 (9.37) ([40.00 - 75.00]) and that for SKOA (N=372) was 61.21 (10.51) ([37.00 - 88.00]). The patients with SKOA and younger than 50 years may have different courses of knee OA. The medical history seemed important for the readers.

2. All three tables used three different subjects groups with different ages. In Table 1, the age range for normal subjects was [40.00 - 75.00], Table 2, [51.00-75.00], and Table 3, [51.00 - 90.00]. Why the subject aged 90 years was not mentioned in Tables 1 and 2.

3. In Table 3, sVAP-1 (ng/ml) –Serum level was not available for normal subjects, but that for OA subjects was 482.5 (132.5) ([180.6 – 695.6]). This important data was missing and may interfere with the interpretation of the data in this study. In addition, the knee OA grading may not be precise or sensitive enough for comparison.

4. Lines 395-406 The authors may need to solve the limitation of inability to compare serum, OA synovial fluid and synovium specimens from the same subjects, before to determine whether local VAP-1 expression and sVAP-1 levels in OA synovium correlate with disease severity and progression. Such a limitation may also need to compare between both knees within the same subjects who had different stages of OA.

5. Regarding the discordance of the serum level and local synovial level of sVAP-1, the authors may also need to assess other joints, e.g., hip, hand, spine, etc, not only knee.

6. Lines 402-405, The knee radiographic data for normal controls used in the synovial fluid study should be studied, because there were some elderly “normal” subjects (age range, [51.00 - 90.00]. The possibility of asymptomatic knee OA should be ruled out before reaching a conclusion to suggest that local VAP-1 expression at early stages of the disease may act to promote and maintain inflammation.

7. Figs. 1 and 2 The magnification of the photos should be presented with a scale bar.